# Observation of the universal magnetoelectric effect in a 3D topological insulator

V. Dziom[1], A. Shuvaev[1], A. Pimenov[1], G.V. Astakhov[2], C. Ames[3], K. Bendias[3], J. Böttcher[4], G. Tkachov[4], E.M. Hankiewicz[4], C. Brüne[3], H. Buhmann[3] & L.W. Molenkamp[3]

The electrodynamics of topological insulators (TIs) is described by modified Maxwell's equations, which contain additional terms that couple an electric field to a magnetization and a magnetic field to a polarization of the medium, such that the coupling coefficient is quantized in odd multiples of $\alpha/4\pi$ per surface. Here we report on the observation of this so-called topological magnetoelectric effect. We use monochromatic terahertz (THz) spectroscopy of TI structures equipped with a semitransparent gate to selectively address surface states. In high external magnetic fields, we observe a universal Faraday rotation angle equal to the fine structure constant $\alpha = e^2/2\epsilon_0 hc$ (in SI units) when a linearly polarized THz radiation of a certain frequency passes through the two surfaces of a strained HgTe 3D TI. These experiments give insight into axion electrodynamics of TIs and may potentially be used for a metrological definition of the three basic physical constants.

[1] Institute of Solid State Physics, Viennta University of Technology, 1040 Vienna, Austria. [2] Physikalisches Institut (EP6), Universität Würzburg, 97074 Würzburg, Germany. [3] Physikalisches Institut (EP3), Universität Würzburg, 97074 Würzburg, Germany. [4] Institut für Theoretische Physik und Astronomie, Universität Würzburg, 97074 Würzburg, Germany. Correspondence and requests for materials should be addressed to A.P. (email: pimenov@ifp.tuwien.ac.at) or to G.V.A. (email: astakhov@physik.uni-wuerzburg.de).

Maxwell's equations are in the foundation of modern optical and electrical technologies. To apply Maxwell's equations in conventional matter, it is necessary to specify constituent relations, describing the polarization $\mathbf{P}_c(\mathbf{E})$ and magnetization $\mathbf{M}_c(\mathbf{B})$ as a function of the applied electric and magnetic fields, respectively. Soon after the theoretical prediction[1–3] and experimental discovery of two-dimensional (2D) and three-dimensional (3D) topological insulators (TIs)[4,5], it has been recognized that the constituent relations in this new phase of quantum matter contain additional cross-terms $\mathbf{P}_t(\mathbf{B})$ and $\mathbf{M}_t(\mathbf{E})$ when time-reversal symmetry is weakly broken[6].

$$\mathbf{P}_t(\mathbf{B}) = \left(N + \frac{1}{2}\right)\frac{\alpha}{2\pi}\mathbf{B}$$
$$\mathbf{M}_t(\mathbf{E}) = -\left(N + \frac{1}{2}\right)\frac{\alpha}{2\pi}\mathbf{E}. \quad (1)$$

Here $N$ is an integer and $\alpha \approx 1/137$ is the fine structure constant. The derivation of equation (1) is based on the topological field theory of time-reversal invariant insulators[6]. Its intriguing consequences are the universal Faraday rotation angle $|\theta_F| = \alpha$, when a linearly polarized electromagnetic radiation passes through the top and bottom topological surfaces[6,7], and magnetic monopole images, induced by electrical charges in proximity to a topological surface[8]. However, experimental verification of these topological magnetoelectric effects (TMEs) has been lacking. As the modified Maxwell's equations describing electrodynamics of TIs are applicable in the low-energy limit, optical experiments should be performed at terahertz (THz) or sub-THz frequencies[9–12]. Qualitatively, equation (1) applied to the magnetic and electric fields of the primary THz radiation results in a perpendicular polarized secondary THz radiation. The sum of the primary and secondary radiation can be viewed as the rotation of the polarization plane, that is, as the Faraday effect. We would like to note that the quantum Faraday effect and the TME are basically different manifestations of the same axion physics[13].

In real samples, the TME may be screened by nontopological contributions[13–15]. In fact, quantized Faraday rotation has been detected in 2D electron gas[16] and graphene[17] in the quantum Hall effect (QHE) regime. However, in both experiments the Faraday rotation takes a value $\theta_F = -4\alpha/(1 + n_{sub})$, that is, it depends on the refractive index of the substrate $n_{sub}$ and hence is not fundamental.

Here we report on the observation of the universal Faraday rotation angle equal to the fine structure constant α. Strained HgTe layers grown on CdTe, that are investigated in the present work, are shown to be a 3D TI[18] with surface-dominated charge transport[19] that was observed in THz experiments as well[9,11,20]. To eliminate the material details, we perform measurements under antireflection conditions, such that the transmission through the CdTe substrate is approaching 100% (ref. 21).

## Results

**THz Faraday effect in 2D systems.** The observed Faraday rotation angle $|\theta_F| = \alpha$ (for $N = 0$) comes from two spatially separated topological surfaces in a 3D TI. This corresponds to the half-quantized Hall conductivity $e^2/(2h)$ per surface or, equivalently, to the TME occurring at each surface separately. Therefore, the observed Faraday effect $2(N + 1/2)\alpha$ is intimately related to the TME, which distinguishes qualitatively our 3D TI from 2D or quasi-2D materials. There is also a quantitative difference. Even without the substrate ($n_{sub} = 1$), the Faraday rotation in graphene would be quantized as $4(N + 1/2)\alpha$, including the spin and valley degeneracies. The minimum Faraday rotation

angle is then $|\theta_F| = 2\alpha$ (for $N = 0$)[17]. For a conventional 2D electron gas, such as hosted in GaAs/AlGaAs heterostructures, the quantization of the Faraday angle is expected to be $2N\alpha$, where the factor of 2 comes from the equal contributions of the up- and down-spin subsystems, which independently exhibit the integer QHE. This is because in GaAs/AlGaAs heterostructures, the Zeeman splitting for magnetic fields below 10 T is negligible compared to the THz photon energy. Therefore, the minimum Faraday rotation angle would be also $|\theta_F| = 2\alpha$ (for $N = 1$)[16], which is twice larger than our result.

**Sample details.** The strained HgTe film is a 58 nm thick HgTe layer embedded between two $Cd_{0.7}Hg_{0.3}Te$ layers (Fig. 1a). The $Cd_{0.7}Hg_{0.3}Te$ layers have a thickness of 51 nm (lower layer) and 11 nm (top/cap layer), respectively. The purpose of these layers is to provide the identical crystalline interface for top and bottom surface of the HgTe films as well as to protect the HgTe from oxidization and adsorption. This leads to an increase in carrier mobility with a simultaneous decrease in carrier density compared to uncaped samples[18]. The transport characterization on a standard Hall bar sample shows a carrier density at 0 V gate of $1.7 \times 10^{11}$ cm$^{-2}$ and a carrier mobility of $2.2 \times 10^5$ cm$^2$ V$^{-1}$ s$^{-1}$. The optical measurements are carried out on a sample fitted with a 110 nm thick multilayer insulator of $SiO_2/Si_3N_4$ and a 4 nm thick Ru film. The Ru film (oxidized in the air) is used as a semitransparent top-gate electrode[22]. The gate leads to $\sim 15\%$ suppression of the transmission signal, which can be taken into account as field-independent contribution to the conductivity. The properties of the gate material have been investigated in a separate experiment.

**THz spectroscopy.** The transmittance experiments at THz frequencies ($0.1$ THz $< \nu < 1$ THz) are carried out in a Mach–Zehnder interferometer arrangement[23,24], allowing measurement of the amplitude and the phase shift of the electromagnetic radiation in a geometry with controlled polarization (Fig. 1a). The monochromatic THz radiation is provided by a backward-wave oscillator. The THz power on the sample is in between 10 and 100 μW with the focal spot of 0.2 cm$^2$. Using wire grid polarizers, the complex transmission coefficient $t = |t|e^{i\phi}$ is obtained both in parallel $t_p$ (Fig. 1b) and cross $t_c$ (Fig. 1c) polarization geometries, providing full information about the transmitted light. External magnetic fields $B \leq 7$ T are applied using a split-coil superconducting magnet. The experiments are carried out in Faraday geometry, that is, with $B$ applied parallel to the propagation direction of the THz radiation. The ac conductivity tensor $\hat{\sigma}(\omega)$ at THz angular frequency $\omega = 2\pi\nu$ is obtained from the experimental data by inverting the Berreman equations[25] for the complex transmission coefficient through a thin conducting film on an insulating substrate. The explicit expressions used in the calculations are given in Methods section.

In general case, the light propagating along the $z$ direction can be characterized by the orthogonal $x$ and $y$ components of the electric and magnetic fields, which can be written in the form of a 4D vector $\mathbf{V}$. The interconnection between vectors $\mathbf{V}_1$ and $\mathbf{V}_2$, corresponding to different points in space separated by a distance $\ell$, is given by $\mathbf{V}_1 = \hat{M}(\ell)\mathbf{V}_2$. Here $\hat{M}(\ell)$ is a $4 \times 4$ transfer matrix. For an insulating substrate of thickness $\ell$ and dielectric constant $\varepsilon$, this is the identity matrix $\hat{M}_{CdTe}(\ell) = \mathbb{I}$ provided $2\ell\sqrt{\varepsilon}\nu/c$ is an integer. We find in a separate experiment on a bare CdTe substrate that this condition is fulfilled for $\nu \approx 0.35$ THz, and all the measurements presented here are performed at this frequency to minimize the contribution to the Faraday signal from the substrate. The corresponding photon energy of 1.4 meV is much

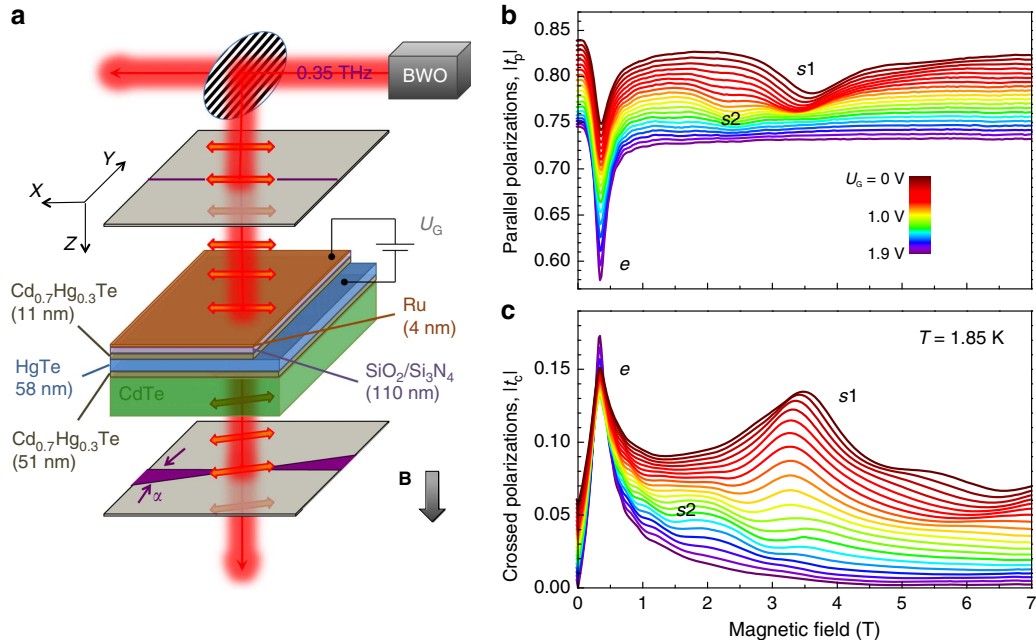

**Figure 1 | THz magnetooptics of a strained HgTe 3D TI.** (**a**) A scheme of the experimental set-up (only one arm of the Mach–Zehnder interferometer is shown). The strained HgTe layer, which is a 3D TI, is sandwiched between (Cd,Hg)Te protecting layers. The top-gate electrode, consisting of a SiO$_2$/Si$_3$N$_4$ multilayer insulator and a thin conducting Ru film, is semitransparent at THz frequencies. The THz radiation ($\nu = 0.35$ THz) is linearly polarized, and the Faraday rotation ($\theta_F$) and ellipticity ($\eta_F$) are measured as a function of the magnetic field strength $B$ for different gate voltages $U_G$. (**b,c**) Transmission spectra in the parallel $|t_p|$ and crossed $|t_c|$ polarizer configurations, respectively. The gate voltage is colour-coded, and the experimental curves are shifted for clarity. Notations in **b,c**: $e$ denotes the CR of the topological surface states of electron character, $s1$ and $s2$ denote extra resonances with opposite phase to that of the $e$-CR as discussed in the text.

smaller than the energy gap in strained HgTe (above 10 meV)[18], and equations (1) are a good approximation.

For normal incidence, the fields across the conducting interface are connected by the Maxwell equation $\nabla \times \mathbf{H} = \hat{\sigma} \mathbf{E}$. Here the $e^{-i\omega t}$ time dependence is assumed for all fields. As the wavelength of 856 µm for $\nu = 0.35$ THz is much larger than the HgTe layer thickness, we use the limit of thin film, and the corresponding transfer matrix $\hat{M}_{\mathrm{HgTe}}(\hat{\sigma})$ is determined by the diagonal ($\sigma_{xx}$) and Hall ($\sigma_{xy}$) components of the conductivity tensor $\hat{\sigma}$. Within the Drude-like model, these components for one type of charge carriers can be written in the form[13,26]

$$\sigma_{xx} = \sigma_{yy} = \frac{1 - i\omega\tau}{(1 - i\omega\tau)^2 + (\Omega_c\tau)^2}\sigma_0, \tag{2}$$

$$\sigma_{xy} = -\sigma_{yx} = \frac{\Omega_c\tau}{(1 - i\omega\tau)^2 + (\Omega_c\tau)^2}\sigma_0. \tag{3}$$

Here $\Omega_c$ is the cyclotron resonance (CR) frequency, $\sigma_0$ is the dc conductivity, and $\tau$ is the scattering time. For classical conductors, the CR frequency is written as $\Omega_c = eB/m_e$, where $m_e$ is the effective electron cyclotron mass.

The total transfer matrix $\hat{M} = \hat{M}_{\mathrm{CdTe}}\hat{M}_{\mathrm{HgTe}}$ relates vectors **V** on both sides of the sample and hence contains full information about the transmission and reflection coefficients. Thus, when $\hat{M}_{\mathrm{CdTe}}$ is the identity matrix, the influence of the substrate is minimized, and the THz response is dominated by the ac transport properties of the HgTe layer, in accord with equations (2) and (3). The calculation of the complex transmission coefficients $t_p$ and $t_c$ based on the transfer matrix formalism as well as the exact form of the transfer matrices are presented in Methods section.

Magnetic field dependence of the THz transmission is dominated by a sharp CR of surface electrons ($e$) $\Omega_{ce}$ at

$B_e = 0.4$ T (Fig. 1b,c). Below we demonstrate their Dirac-like character and that they are responsible for the universal Faraday rotation. Remarkably, the observation of the CR both in $t_p$ and $t_c$ indicates a high purity of our HgTe layer. The scattering time is significantly longer than the inverse THz frequency $\omega\tau \gg 1$, and according to equations (2) and (3) the ac conductivity reveals a resonance-like behaviour $\sigma_{xx}, \sigma_{xy} \propto 1/(\Omega_{ce}^2 - \omega^2)$.

Further features are broad resonances at $B_{s1} = 3.7$ T and at $B_{s2} = 2.2$ T indicated in Fig. 1 as $s1$ and $s2$, respectively. The phase of the corresponding THz transmission coefficient $\phi_c$ in the vicinity of these resonances has the opposite sign with respect to that of the $e$-CR. Remarkably, the $s1$ and $s2$ resonances disappear with applying positive gate voltage (Figs 1 and 2b). We associate them with either interband Landau-level transitions or thermally activated states as discussed below.

**Band structure analysis.** To understand the origin of the experimentally observed resonances, we analyse the band structure of tensile strained Cd$_{0.7}$Hg$_{0.3}$Te/HgTe layer as shown in Fig. 2a. It is obtained similar to ref. 19 within the tight binding approximation of the 6 × 6—Kane Hamiltonian[27,28]. Due to reduced point symmetry at the boundary between the Cd$_{0.7}$Hg$_{0.3}$Te and HgTe layers, an additional interface potential is allowed in the Hamiltonian[29]. This potential is used to shift the Dirac point closer to the valence band edge, so that the tight binding results are in good agreement with recent angle-resolved photoemission spectroscopy (ARPES) experiments[18,30] and ab initio calculations[31] on HgTe. Final position of the Dirac point is at energy around $-40$ meV (that is, it is buried in the the valence band) in agreement with[18,30,31]. The Dirac-like surface states are located in the band gap between the light-hole (LH, conduction) and heavy-hole (HH, valence) subbands (see the red

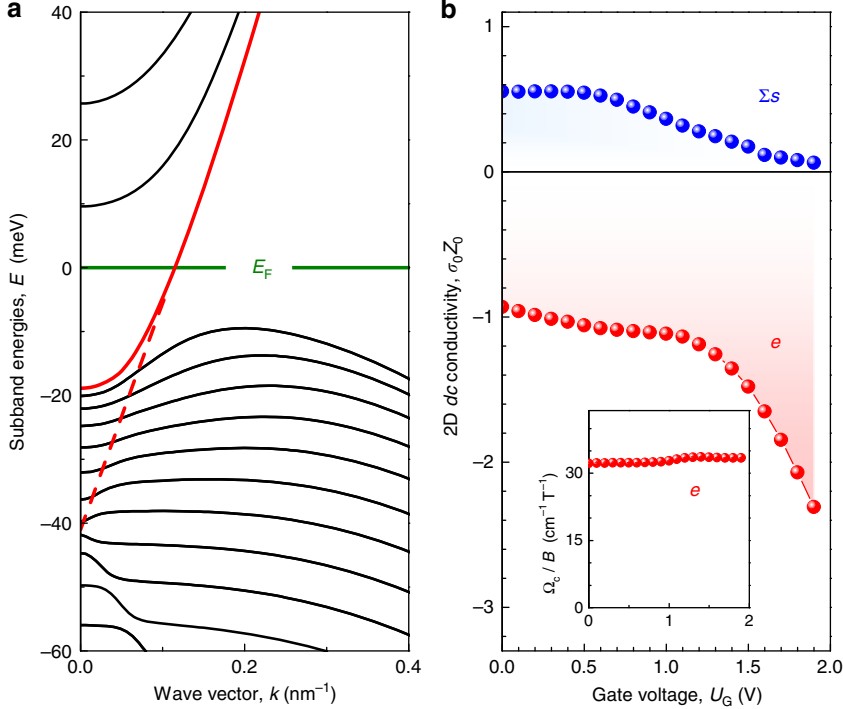

**Figure 2 | Charge carriers in strained HgTe.** (a) The band structure of the $Cd_{0.7}Hg_{0.3}Te/HgTe$ heterostructure close to the $\Gamma$-point for $U_G = 1.9$ V. The chemical potential (green line) crosses the Dirac-like surface state (red solid line) in the band gap corresponding to the electron CR $\Omega_{ce}$. The dashed red line shows the linear dispersion of the 2D surface state without hybridization with the heavy-hole band. (b) 2D dc conductivity $\sigma_0$ of different charge carriers ($e$ and $\Sigma s = s1 + s2$), obtained by Drude-like fits to equations (2) and (3) of the magnetooptical spectra. The dimensionless values are given relative to the impedance of free space $Z_0 = 1/c\epsilon_0 \approx 377$ Ohm. The inset shows the $e$-CR in terms of $\Omega_{ce}/B$ as a function of the gate voltage $U_G$.

line in Fig. 2a), and they are not perfectly linear due to the hybridization with the heavy-hole band. The dashed red line in Fig. 2a shows the linear dispersion of the 2D surface state without hybridization with the heavy-hole band. This is a hypothetical curve, since in the realistic material the hybridization with the heavy-hole band is non-zero. The camel back of the heavy-hole band originates from coupling of this band to the electron-like valence band and is therefore a hallmark of the inverted band structure of HgTe. In accordance with previous transport data, the chemical potential crosses the topological surface states for a large range of gate voltages[19]. The total electron density in Fig. 2a is $n \approx 2\times10^{11}$ cm$^{-2}$, representing the experimental situation at $U_G = 1.9$ V. For simplicity, we assume here the same density at the top and bottom surfaces. Using the general formula for a quasi-classical CR[32] $\Omega_c = \frac{2\pi eB}{\hbar^2}\frac{\partial E(k)}{\partial A}$, where $E(k)$ is the energy dispersion, $B$ is the magnetic field and $A$ is the area enclosed by the wave vector $k$, we calculate for the topological surface state $\Omega_{ce}/B \approx 35$ cm$^{-1}$T$^{-1}$.

**Quantized THz Hall effect.** Experimentally, simultaneous fit of the real and imaginary parts of $t_p$ and $t_c$ allows the extraction of all transport characteristics, that is, conductivity, charge carrier density, scattering time and CR frequency[21]. The inset of Fig. 2b shows experimentally determined electron CR as a function of gate voltage, which perfectly agrees with the theoretical value for the topological Dirac-like surface states. Since only surface states are observed in transport experiments on the similar structures[19], a possible explanation of the appearance of additional resonances is interband Landau-level transitions between the HH bulk bands and topological surface states. Such transitions are generally allowed as can be shown

using the Kubo formula. Another possibility would be thermally activated transport between the camel back of the HH bulk band and the surface states. This is generally possible since the THz field may well induce heating of the carriers, resulting in a higher effective temperature compared to that of the lattice.

From the obtained scattering time and the CR positions in the magnetooptical spectra of Fig. 1b,c, one can calculate the mobility $\mu = \tau\Omega_c/B$. The surface states demonstrate high mobility $\mu_e = 1.8\times10^5$ cm$^2$ V$^{-1}$ s$^{-1}$, which agrees with the dc transport data. Since the $e$-CR and $s1,s2$ resonances occur at different magnetic fields, their contributions to the ac transport can be clearly separated, as presented in Fig. 2b. The striking feature of this plot is that the ac conductivity of the surface states dominates at large gate voltages. In what follows, we concentrate therefore on $U_G > 1.0$ V, while remaining weak contribution from the interband Landau-level transitions/thermally activated transitions are subtracted as explained in Supplementary Fig. 1.

Figure 3a demonstrates the real part of the Hall conductivity $\sigma_{xy}$. The overall behaviour is provided by the high-field tail of the classical Drude model, that is, equation (3), resulting in a rapid suppression of $\sigma_{xy}$ with growing magnetic field. In addition to the classical behaviour, regular oscillations in $\partial\sigma_{xy}/\partial B$ can be recognized, which are linear in inverse magnetic field (Fig. 3b). The slope of the linear behaviour changes with gate voltage, reflecting gate dependence of the electron density per surface. These QHE oscillations are extrapolated to 1/2, indicating Dirac character of the surface electrons[33]. The oscillations of $\partial\sigma_{xy}/\partial B$ in Fig. 3a are superimposed by the 1/B tail of the classical electron CR and therefore are not well pronounced. The visibility can be significantly improved by inserting the sample in a Fabry–Pérot resonator, as we have previously demonstrated for a similar structure[11].

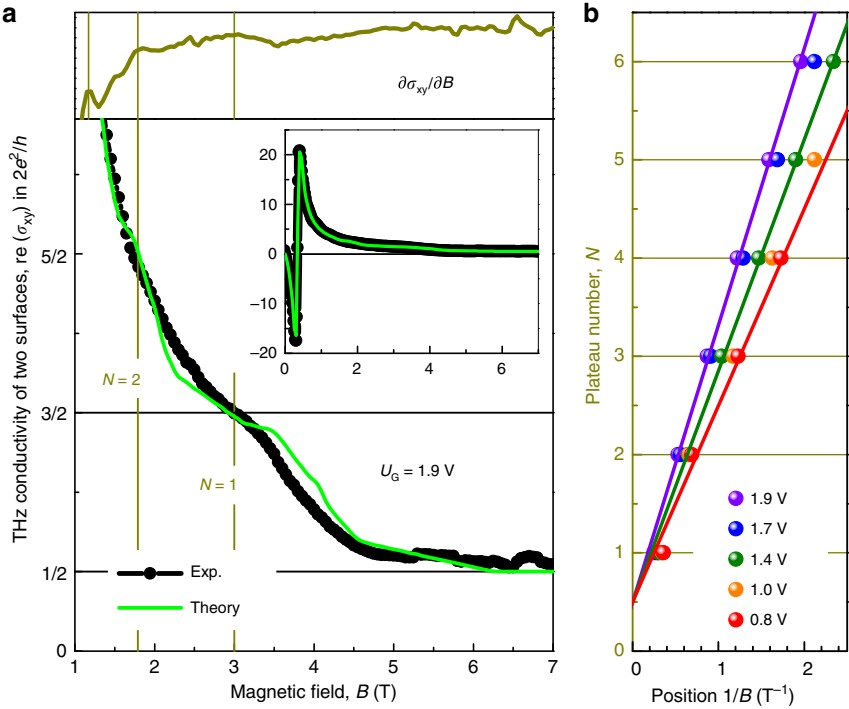

**Figure 3 | THz QHE of the surface states.** (**a**) The real part of the THz Hall conductivity $\sigma_{xy}$ of two surfaces in units of $2e^2/h$, obtained at $U_G = 1.9$ V (symbols). The vertical solid lines indicate the positions of the Hall plateaus, estimated from the maxima in $\partial\sigma_{xy}/\partial B$. Theoretical calculations represented by the thin line are performed as explained in the text. The inset presents the same experimental and theoretical curves in the whole magnetic field range, including the surface carrier CR at 0.4 T. (**b**) The plateau number as a function of the inverse magnetic field for different gate voltages. The lines are linear fits to $N = hn_a/eB + 1/2$, where $n_a$ corresponds to the carrier density on a single surface.

In magnetic fields above 5 T, the Hall conductivity clearly shows a plateau close to $\sigma_{xy} = e^2/h$, corresponding to a value $(1/2)e^2/h$ per surface (Fig. 3a). Another plateau close to $(3/2)e^2/h$ per surface is also recognizable at a magnetic field of 3 T. It is superimposed on the CR tail and therefore tilted. The steps in $\sigma_{xy}$ loose their regularity in lower-magnetic fields, as can be qualitatively explained by a finite THz frequency $\omega$ in magnetooptical experiments. As mentioned above, the overall behaviour of $\sigma_{xy}(\omega)$ is provided by the classical curve of equation (3), and the real part of $\sigma_{xy}$ can be approximated as $\sigma_{xy}(\omega) \approx \sigma_0 \Omega_{ce}/[(\Omega_{ce}^2 - \omega^2)\tau]$, which in the limit $\Omega_{ce} \gg \omega$ reduces to the expression $\sigma_{xy} = ne/B$, being a multiple of $e^2/h$. In low magnetic fields, the CR frequency $\Omega_{ce}$ becomes comparable to the THz frequency $\omega$, destroying the regularities in $\sigma_{xy}(\omega)$.

Since in strained HgTe, the Fermi level lies in the bulk band gap (see Fig. 2a and ref. 19), we attribute the observed THz QHE to the formation of the 2D Landau levels at the top and bottom surfaces of the HgTe layer (Fig. 1a). This interpretation is further substantiated by our theoretical analysis of the *ac* quantum Hall conductivity $\sigma_{xy}(\omega)$ calculated from the Kubo formula for both top and bottom surface states within the Dirac model[13,34].

Our two-surface Dirac model describes well the surface carrier CR (the inset of Fig. 3a). The lengths of the theoretical Hall plateaus in the high magnetic field region (Fig. 3a) correlate correctly with the positions of the extrema in the derivative $\partial\sigma_{xy}/\partial B$. However, the model predicts much sharper transitions between the QHE plateaus, as observed in the experiment. One of two possible explanations is the heating of the surface carriers by the THz field, resulting in a higher effective temperature compared to that of the lattice. Such a heating can occur due to inefficient energy relaxation in the electronic system through the emission of LO phonons at low temperatures[35]. The best fit

of our experimental data is obtained with $T = 25$ K (Fig. 3a). Another explanation is based on spatial fluctuations of the surface carrier densities, which are likely to occur in our samples due to their large lateral sizes compared to the typical Hall bars used in the *dc* measurements. The experimental data of Fig. 3a can alternatively be well fitted assuming cold carriers ($T = 1.8$ K) with density fluctuations within 10% relative to their nominal values. As the fits are nearly indistinguishable, we cannot quantitatively determine the contributions of both mechanisms leading to the smearing of the THz QHE plateaus.

The stronger the field, the closer the Hall conductivity to the quantized values expected for a two-surface Dirac system

$$\sigma_{xy} = (N_a + N_b + 1)\frac{e^2}{h}, \quad N_{a,b} = \text{Int}\left(\frac{n_{a,b}\Phi_0}{|B|}\right), \quad (4)$$

where $N_{a,b}$ are the integer numbers of the highest occupied Landau levels at the top and bottom surfaces, with $n_a \approx n_b$ being the corresponding carrier densities ($\Phi_0 = h/|e|$ is the magnetic flux quantum). Upon approaching the CR, the Hall conductivity deviates from the quantized values in equation (4) due to the predominance of the intraband transitions between the Landau levels. From the fitting procedure, we extract the nominal carrier densities $n_a = 0.92 \times 10^{11}$ cm$^{-2}$ and $n_b = 1.07 \times 10^{11}$ cm$^{-2}$. The total surface carrier density $n_a + n_b$ agrees well with that obtained from Drude-like fits of magnetooptical spectra. Another fitting parameter is the classical (Drude) surface conductivity $\sigma_a = \sigma_b \approx 50e^2/h$. Its large value indicates high-surface carrier mobility, insuring that the condition for the quantum Hall regime, $\sqrt{2}\Omega_B\tau_{a,b} = 4R_0\sigma_{a,b}\sqrt{(|B|/n_{a,b}\Phi_0)} > 1$, is met for $B > 1$ T. Here $\sqrt{2}\hbar\Omega_B = v_F(2\hbar|eB|)^{1/2}$ is the characteristic Landau-level spacing for a Dirac system, $\tau_{a,b}$ are the scattering times of the top

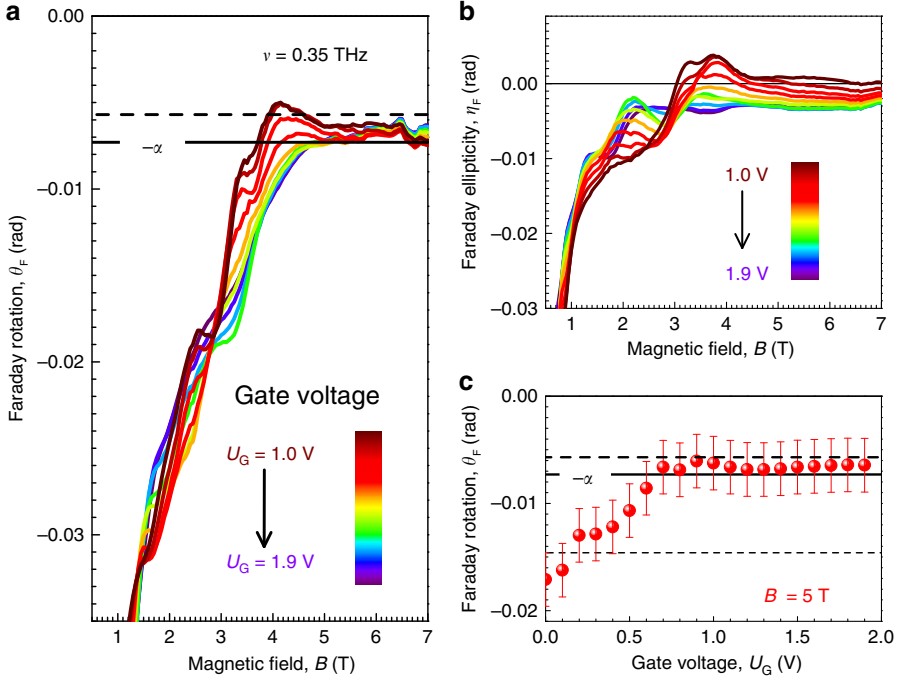

**Figure 4 | Quantized THz Faraday rotation of Dirac fermions.** Faraday rotation (**a**) and Faraday ellipticity (**b**) in a 3D HgTe TI as a function of the external magnetic field for different gate voltages (colour-coded). The horizontal solid line in **a** indicates the universal Faraday rotation angle $\theta_F = -\alpha \approx -7.3 \times 10^{-3}$ rad. (**c**) Gate voltage dependence of the Faraday rotation in a magnetic field of 5 T. The dashed lines in **a,c** provide the calculated value of the Faraday rotation with realistic parameters of the experiment (including the top-gate electode) and assuming the HgTe conductivity exactly equal to $\sigma_{xy} = e^2/h$. The short-dashed line in **c** indicates the $\theta_F = -2\alpha$ level. The error bars are estimated taking into account the accuracy of the original transmission data and the uncertainties due to subtraction of weak residual contribution (Supplementary Fig. 1).

and bottom carriers, $R_0 = h/(2e^2)$ is the resistance quantum, and $v_F$ is the Fermi velocity of the Dirac surface states.

**Quantized THz Faraday effect.** Having established that the THz response of the topological surface states in high magnetic fields $B > 5$ T is determined by the conductivity quantum $G_0 = e^2/h$ ($N_a = N_b = 0$), we turn to the central result of this work, the THz Faraday effect. Owing to the TME of equation (1), an oscillating electric $E_x e^{-i\omega t}$ (magnetic $H_y e^{-i\omega t}$) field of the linearly polarized THz radiation induces in a 3D TI an oscillating magnetic $\alpha H_x e^{-i\omega t}$ (electric $\alpha E_y e^{-i\omega t}$) field. The generated, in such a way, secondary THz radiation is polarized perpendicular to the primary polarization and its amplitude is $\alpha$ times smaller. This can be viewed as a rotation of the initial polarization by an angle $|\theta_F| = \arctan\alpha \approx 7.3 \times 10^{-3}$ rad. Indeed, Fig. 4a clearly demonstrates that the Faraday angle in high magnetic fields is close to this fundamental value.

We rigorously characterize the THz Faraday effect, and the Faraday ellipticity $\eta_F$ is shown in Fig. 4b. It is relatively small $|\eta_F| < |\theta_F|$ in high magnetic fields, but does not reach zero. This observation indicates that while the TME dominates, the interaction of TIs with THz radiation is not a completely dissipationless process in our samples. Remarkably, the universal value of the Faraday angle remains robust against the gate voltage. This is demonstrated in Fig. 4c, where $|\theta_F| \approx \alpha$ for $0.7$ V $< U_G < 1.9$ V.

**Discussion**

The observed terahertz Faraday rotation equal to the fine structure constant $\alpha = e^2/2\epsilon_0 hc = e^2\mu_0 c/2h$ is a direct consequence of the TME, confirming axion electrodynamics of 3D topological insulators. We use monochromatic terahertz spectroscopy,

providing complete amplitude and phase reconstruction, which can be applied to investigate topological phenomena in various systems, including graphene, 2D electron gas, layered superconductors and recently experimentally discovered Weyl semimetals[36]. Picoradian angle resolution can be achieved using a balanced detection scheme[37], and the universal Faraday rotation in combination with the magnetic flux quantum $\Phi_0 = h/|e|$ and the conductivity quantum $G_0 = e^2/h$ are suggested[14] to use for a metrological definition of the three basic physical constants, $e$, $h$ and $c$ (given that the vacuum permeability is equal exactly $\mu_0 = 4\pi \times 10^{-7}$ N/A$^2$).

*Note added in proof*: Okada *et al.* and Wu *et al.* reported on the quantized Faraday rotation in TIs independently to this work and around a similar time in arXiv. We note that Okada *et al.*[38] observed only the trajectory towards the fine structure constant. Wu *et al.*[39] used interface doping to reduce the carrier concentration and put the chemical potential in the bulk.

**Methods**

**Experimental technique.** Magnetooptical experiments in the THz frequency range (100 GHz $< v <$ 1,000 GHz) have been carried out in a Mach–Zehnder interferometer arrangement[23], which allows measurements of the amplitude and the phase shift in a geometry with controlled polarization of radiation.

**Theoretical analysis of magnetooptical spectra.** To analyse the experimental transmission spectra, we follow the formalism described by Berreman[24,25].
The THz light propagating along the $z$ direction can be characterized by the (orthogonal) components of electric ($E_x$, $E_y$) and magnetic ($H_x$, $H_y$) fields, which may be combined in form of a four-component vector $\mathbf{V} = (E_x, E_y, H_x, H_y)$. The propagation of light between two points in space separated by a distance $\ell$ and characterized by vectors $\mathbf{V}_1$ and $\mathbf{V}_2$ can be described via $4 \times 4$ transfer matrix $\hat{M}(\ell)$ as $\mathbf{V}_2 = \hat{M}(\ell)\mathbf{V}_1$. To provide a simple example, for an isotropic dielectric substrate

the transfer matrix is simplified to:

$$\hat{M}_{CdTe}(\ell) = \begin{pmatrix} \cos(k\ell) & 0 & 0 & iZ\sin(k\ell) \\ 0 & \cos(k\ell) & -iZ\sin(k\ell) & 0 \\ 0 & -iZ^{-1}\sin(k\ell) & \cos(k\ell) & 0 \\ iZ^{-1}\sin(k\ell) & 0 & 0 & \cos(k\ell) \end{pmatrix}. \quad (5)$$

Here $Z = \sqrt{\mu/\varepsilon}$ and $k = \sqrt{\mu\varepsilon}\,\omega/c$ is the wave vector. In the following, we assume $\mu = 1$. The Berreman procedure is in general not limited to the case of normal incidence. However, in such geometry the choice of tangential field components simplifies the treatment. Electric and magnetic fields across the interfaces are connected by the Maxwell equation $\nabla \times \mathbf{H} = \hat{\sigma}\mathbf{E}$. Here $\hat{\sigma}$ is the complex conductivity tensor of the material and the time dependence in form $e^{-i\omega t}$ is assumed.

The full transfer matrix $\hat{M} = \hat{M}_{CdTe}\hat{M}_{HgTe}$ describes the transmission and reflection coefficients, which can be calculated using another basis. In this basis, the vector $\mathbf{V}$ consists of (i) the amplitude of the linearly polarized wave ($E_x$) propagating in the positive direction, (ii) the amplitude of the wave with the same polarization propagating in the negative direction, and (iii) and (iv) of two waves with perpendicular polarization ($E_y$). The propagation matrix in the new basis is $\hat{M}' = \hat{V}^{-1}\hat{M}\hat{V}$, with

$$\hat{V} = \begin{pmatrix} 1 & 1 & 0 & 0 \\ 0 & 0 & 1 & 1 \\ 0 & 0 & -1 & 1 \\ 1 & -1 & 0 & 0 \end{pmatrix}. \quad (6)$$

The present experiment is described by a linearly polarized incident wave and by two components of the transmitted ($t$) and reflected ($r$) waves, respectively. The equation connecting all waves is given by:

$$\begin{pmatrix} t_c \\ 0 \\ t_p \\ 0 \end{pmatrix} = \hat{M}' \begin{pmatrix} 0 \\ r_c \\ 1 \\ r_p \end{pmatrix}. \quad (7)$$

Here the $t_p$ and $t_c$ are the complex transmittance amplitudes within parallel and crossed polarizers geometry, $r_p$ and $r_c$ are respective reflectivity coefficients. The Faraday rotation $\theta$ and Faraday ellipticity $\eta$ are obtained from the transmission amplitudes $|t_p|$, $|t_c|$ and phase shifts $\phi_p$, $\phi_c$ as

$$\tan(2\theta) = \frac{2|t_p||t_c|\cos(\phi_p - \phi_c)}{|t_p|^2 - |t_c|^2},$$
$$\sin(2\eta) = \frac{2|t_p||t_c|\sin(\phi_p - \phi_c)}{|t_p|^2 + |t_c|^2}. \quad (8)$$

To interpret the experimental data, we use the $ac$ conductivity tensor $\hat{\sigma}(\omega)$ obtained in the classical (Drude) limit from the Kubo conductivity of topological surface states[7,15]. We note that for a 2D conducting film on an isotropic dielectric substrate, the complex transmission can be obtained analytically. For a substrate with permittivity $\varepsilon$, the final equations for the spectra in parallel ($t_p$) and crossed ($t_c$) polarizers are given by:

$$t_p = \frac{2a_{xx}}{a_{xx}^2 + a_{xy}^2}, \quad (9)$$

$$t_c = \frac{2a_{xy}}{a_{xx}^2 + a_{xy}^2}, \quad (10)$$

where

$$a_{xx} = (1 + \sigma_{xx}Z_0)(\cos(k\ell) - iZ\sin(k\ell)) + (\cos(k\ell) - iZ^{-1}\sin(k\ell)), \quad (11)$$

$$a_{xy} = \sigma_{xy}Z_0(\cos(k\ell) - iZ\sin(k\ell)). \quad (12)$$

Here $\ell$ is equal to the substrate thickness, $Z_0 \approx 377$ Ohm is the impedance of free space, $Z = 1/\sqrt{\varepsilon}$ is the relative impedance of the substrate and $k = \sqrt{\varepsilon}\omega/c$ is the wave vector of the radiation in the substrate. The components of the conductivity tensor, $\sigma_{xx}(\omega)$, and $\sigma_{xy}(\omega)$, are given by equations (2) and (3). As can be clearly seen, equations (9) and (10) can be inverted analytically to get the solution for the complex conductivity thus avoiding the numerical procedure.

As has been discussed previously[6], in case of a film on a substrate the universal value of the Faraday rotation angle should be modified by the refractive index of the substrate. We note that an infinite substrate has been assumed in these calculations. In the present case of finite substrate, exact transmission matrix formalism has been utilized thus automatically taking into account the influence of the substrate and of the Ru-gate. Moreover, monochromatic radiation has been used in the experiments and the frequency of this radiation has been selected close to a maximum of the Fabry–Pérot resonances in the substrate ($\sin(k\ell) \to 0$). In this case, equations (11) and (12) can be simplified to

$$a_{xx} = 2 + \sigma_{xx}Z_0 \quad (13)$$

$$a_{xy} = \sigma_{xy}Z_0 \quad (14)$$

and the influence of the substrate is minimized.

**Data availability.** The data that support the findings of this study are available from the corresponding authors upon reasonable request.

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

## Acknowledgements

This work was supported by Austrian Science Funds (I1648-N27, W-1243, P27098-N27), as well as by the German Research Foundation (DFG) through SFB 1170 'ToCoTronics', the SPP 1666 and TK60/4-1, the Bavarian ENB Graduate School on Topological Insulators and the ERC (AG project 3-TOP).

## Author contributions

V.D., A.S. and A.P. contributed to the THz experiments. C.A., K.B., C.B. and H.B. grew the samples and fabricated the gate electrodes. J.B., G.T. and E.M.H. performed the theoretical analysis. A.S., A.P., G.V.A., C.B., H.B. and L.W.M. conceived the experiment. G.V.A., A.P. and L.W.M. wrote the experimental part of the paper. G.T. and E.M.H. wrote the theoretical part of the paper. G.V.A. coordinated the research project. All authors participated in the interpretation of the experiments.

## Additional information

**Competing interests:** The authors declare no competing financial interests.

**Publisher's note**: 

