## [Peer Review File · Nature Communications]

Reviewers' comments:

Reviewer #1 (Remarks to the Author):

The manuscript "Observation of the universal magnetoelectric effect in a 3D topological insulator" by V. Dziom and co-authors reports about magneto-optical study of a so-called 3D topological insulator. It is claimed that when a radiation passes through two surfaces of a strained HgTe, the Faraday rotation becomes quantized and from this quantization one can define the fine structure constant. I think that this is a new and interesting result. Apart from an obvious impact on the society specialized on the study of topological materials, I am sure that also specialists in magneto-optical spectroscopy, magneto-electricity and probably multiferroics will find this work interesting. At least I, being intrigued by the introduction, read this paper with interest mainly from the point of view of magneto-electricity and magneto-photonics.

To perform this study the authors measured the Faraday rotation and ellipticity in the THz spectral range and in the fields up to 7 Tesla. For the measurements the authors used specially designed structures with a stress and anti-reflection coatings of the studied layer. The experiment also allowed to apply electric fields to the studied sample. In short, the paper report about very original experiments and the experimental data are of very high quality. Although the conclusions are, in principle, I also have a couple of comments/concerns.

This is mainly related to the fact that I am not sure if the authors really need to call the observed phenomenon as a magneto-electric effect. The problems start with Eqs.1 Right and left parts of the equation have different symmetry with respect to time-reversal. Normally it means that such phenomena can be observed only in magnetically ordered media.

Next I was wondering if B is the magnetic field applied in the experiment by an electromagnet (it seems so), or it is the sum of the field applied in the experiment plus and the field of the electromagnetic wave. However, already in the next paragraph instead of discussing the magneto-electric effect the authors start to talk about the Faraday rotation. The rest of the manuscript also discusses the Faraday effect in terms of off-diagonal terms of electric conductivity. I am convinced by the detailed analysis of the observed Faraday rotation, but the Faraday and magneto-electric effect are two totally different phenomena. The link suggested by the authors between the actually observed Faraday effect and the magneto-electric effect claimed by the title (see also Eq. 1) is not obvious to me.

To summarize, I think this is an interesting article. However, the title and the introduction seem to be misleading to me. I would be happy if this paper is published by Nature Communications, but first the authors should clarify the issues mentioned above, improve the paper and by making it less more accessible for a non-specialists.

Reviewer #2 (Remarks to the Author):

This MS reports on the observation of a quantized Faraday effect in a measurement of infrared light transmission through a 3D topological insulator. The substrate thickness has been carefully chosen to minimize its role in the measurement. The work is extremely interesting and I believe that it should be published.

There are no technical problems with this MS - but there is a bit of controversy surrounding the language the authors use to describe their results. I have suggestions on how this politics can be navigated in a way that serves the interests of readers. There are two main elements of the politics.

i) It is claimed theoretically that 3D topological insulators are magneto-electric materials with a quantized coupling coefficient. This claim is equivalent to saying that the materials have a surface quantum Hall effect that is 'half-quantized'. In ideal samples this claim is valid when time-reversal symmetry is infinitesimally broken. In real samples, the claim is valid only for sufficiently strong time reversal symmetry breaking. The experiments reported on here are performed in a magnetic field that is 'sufficiently strong'. To acknowledge this important point I recommend that the authors add the words 'when time-reversal symmetry is weakly broken' before 'the constituent relations' above Eq.1.

ii) The experiment that is described measures the sum of the Hall conductivities of the top and bottom surfaces of the 2D materials - and not the surfaces individually. The current discussion at the bottom of page 2 and the top of page 3 does a good job of explaining the similarities and differences relative to other quasi-2D systems in an external magnetic field.

Reviewer #3 (Remarks to the Author):

Dziom et al., reported universal Faraday rotation derived from the quantized Hall effect on the topological insulator film by the monochromatic terahertz spectroscopy. On the surface of the topological insulator, the novel magnetoelectric coupling derived from the axionic term in Lagrangian formalism is expected to show the universal phenomena including quantized Faraday rotation. They measured the terahertz Faraday rotation at fixed frequency of 0.35 THz, where the contribution from the substrate effectively vanishes, resulting in the rotation angle of free standing film. The HgTe film shows the quantized Hall effect under the magnetic field and the Fermi level can be tuned by the gate voltage. The observed rotation angle shows the convergence towards the quantized value, which is equal to the fine structure constant.

They claimed the possible metrological definition of the fine structure constant by rotation angle at terahertz frequency, however, other quantized Hall system also show the quantized rotation of polarization of light, which is an integer multiple of the fine structure constant, as they mentioned in the text. The lower-lying tail of the cyclotron resonance inevitably affects the rotation angle, which was actually observed in their measurement. So I don't agree the advantage on this point.

In the text, authors said "experimental verification of these TMEs has been lacking." in the introductory paragraph. But similar results by other two groups were uploaded on the online archive in March, in which the quantized Faraday rotations on topological insulator film by terahertz spectroscopy have been reported (arXiv:1603.02113 and arXiv:1603.04317v3) in addition to the authors' work.

One of them has been already published in the nature communications in July (Nat. Commun. 7:12245). These works should be mentioned in the main text.

I admit the technical advantages for the use of the gate, which the other groups didn't apply, and also for the use of the "magic frequency", which yields the optical responses of free standing film sample. However, the novelty and the impact of this research is limited, so I don't recommend the paper for publication on Nature communications.

I list my concerns below.

1. Authors displayed only the optical data except for Fig. 2. I think the characterization of their film sample with the optically transparent gate is necessary. The rotation angle in terahertz frequency can be scaled by the evolution of the quantization in dc limit, so σ_{xy} at dc with the deduced rotation

angles should be compared with optical data in Fig. 3 and 4.

2. Topological magnetoelectric effect is ascribed to the origin of the Faraday rotation. However, the quantized Hall effects always lead to the quantized Faraday rotation, even in the general quantum Hall effect. Could the authors distinguish the TMEs from conventional quantum Hall effects in 2D systems.

3. Temperatures of these experiments should be noted in all Figures.

4. Authors assumed the heating of the electronic system. The power of terahertz beam is several tenth micro W, it seems that the energy flux is low enough to avoid the heating to 25 K. In addition, the transparency in Fig. 1 is high.

REVIEWERS' COMMENTS:

Reviewer #1 (Remarks to the Author):

The authors have fully addressed my criticism. From the reports of other Referees and the response of the authors I do not see any reasons which may prevent the publication.

Reviewer #2 (Remarks to the Author):

The so-called 'topological magneto-electric effect' is different from the quantum Faraday effect - but only in a quantitative way. I think that with the revisions present in the revised version of the MS the authors do an adequate job of navigating the terminology - which is somewhat loaded.

The experiment is clearly described and seems solid. There is no technical issue that I can see.

Reviewer #3 (Remarks to the Author):

The authors have improved their manuscript, but I am not convinced that it is sufficiently innovative to merit publication in Nature communications. At least in the finite magnetic field, the rotation angle is inevitably suffered from the tail of the classical cyclotron resonance, even if their contribution decreases in the high magnetic field. They concluded that the observed rotation angle is "equal to" the fine structure constant, whereas the error bar (~ 4 mrad) is as large as the quantized rotation angle (~ 7.3 mrad) (Fig. 4c). The current results are still far from the metrological definition of the fine structure constant, while I agree that the observed Faraday rotation indicates the TME. Two groups have already published the similar results in Science and Nature communications, which also indicate the TME, so that the justification for the publication will be necessary. I think the merits of this work are high quality HgTe film and the terahertz spectroscopy with use of gate control. But, in my opinion, these merits do not meet the criteria of the nature communications.

General Response

We thank the referees for their constructive comments and remarks, which help us to make the presentation of our results clearer for readers. We have answered all the question and added corresponding comments/citations in the manuscript. We hope that our manuscript is now suitable for publication in the present form.

Response to Reviewer #1

Comment 1: "This is mainly related to the fact that I am not sure if the authors really need to call the observed phenomenon as a magneto-electric effect. The problems start with Eqs.1 Right and left parts of the equation have different symmetry with respect to time-reversal. Normally it means that such phenomena can be observed only in magnetically ordered media."

Such phenomena can be also observed in topological surface states due to an additional $\mathbf{E}\cdot\mathbf{B}$ term in the Maxwell Lagrangian. The topological field theory of time-reversal invariant insulators is described in Ref. 6 in great detail. We now emphasize this more clearly.

Comment 2: "Next I was wondering if B is the magnetic field applied in the experiment by an electromagnet (it seems so), or it is the sum of the field applied in the experiment plus and the field of the electromagnetic wave. However, already in the next paragraph instead of discussing the magneto-electric effect the authors start to talk about the Faraday rotation. The rest of the manuscript also discusses the Faraday effect in terms of off-diagonal terms of electric conductivity. I am convinced by the detailed analysis of the observed Faraday rotation, but the Faraday and magneto-electric effect are two totally different phenomena. The link suggested by the authors between the actually observed Faraday effect and the magneto-electric effect claimed by the title (see also Eq. 1) is not obvious to me."

Eqs. (1) applied to the magnetic and electric fields of the primary THz radiation results in a perpendicular polarized secondary THz radiation. The sum of the primary and secondary radiation can be viewed as the rotation of the polarization plane, i.e., as the Faraday effect. We would like to note that the quantum Faraday effect and the topological magnetoelectric effect are basically different manifestations of the same axion physics, as clearly explained by Tse and MacDonald (Ref. 14). We have added this discussion after Eq. (1).

Response to Reviewer #2

Comment 1: "It is claimed theoretically that 3D topological insulators are magneto-electric materials with a quantized coupling coefficient. This claim is equivalent to saying that the materials have a surface quantum Hall effect that is 'half-quantized'. In ideal samples this claim is valid when time-reversal symmetry is infinitesimally broken. In real samples, the claim is valid only for sufficiently strong time reversal symmetry breaking. The experiments reported on here are performed in a magnetic field that is 'sufficiently strong'. To acknowledge this important point I recommend that the authors add the words 'when time-reversal symmetry is weakly broken' before 'the constituent relations' above Eq.1."

We have added this phrase, as recommended by the reviewer.

Response to Reviewer #3

General comment: "They claimed the possible metrological definition of the fine structure constant by rotation angle at terahertz frequency, however, other quantized Hall system also show the quantized rotation of polarization of light, which is an integer multiple of the fine structure constant, as they mentioned in the text. The lower-lying tail of the cyclotron resonance inevitably affects the rotation angle, which was actually observed in their measurement. So I don't agree the advantage on this point."

We disagree with this statement. The classical contribution from the cyclotron resonance to the Faraday rotation decays in strong magnetic fields, asymptotically follow $1/B^2$, and therefore can be suppressed. In contrast, the quantum Faraday effect remains equal to the fine structure constant.

General comment: "But similar results by other two groups were uploaded on the online archive in March, in which the quantized Faraday rotations on topological insulator film by terahertz spectroscopy have been reported (arXiv:1603.02113 and arXiv:1603.04317v3) in addition to the authors' work. One of them has been already published in the nature communications in July (Nat. Commun. 7:12245). These works should be mentioned in the main text."

We now cite these works.

Comment 1: "Authors displayed only the optical data except for Fig. 2. I think the characterization of their film sample with the optically transparent gate is necessary. The rotation angle in terahertz frequency can be scaled by the evolution of the quantization in dc limit, so σ_{xy} at dc with the deduced rotation angles should be compared with optical data in Fig. 3 and 4."

There were no Hall bars fabricated on the top of our samples used for the THz experiments, as their typical size is much less than the wavelength corresponding to 0.35 THz. DC experiments on a similar sample are discussed in detail in Ref. 19.

Comment 2: "Topological magnetoelectric effect is ascribed to the origin of the Faraday rotation. However, the quantized Hall effects always lead to the quantized Faraday rotation, even in the general quantum Hall effect. Could the authors distinguish the TMEs from conventional quantum Hall effects in 2D systems."

The Reviewer would like to see difference from conventional QHE. The total THz conductivity presented in Fig. 3 shows the $\sigma_{xy} = e^2/h$ ($N = 0$) and $\sigma_{xy} = 3e^2/h$ ($N = 1$) Hall plateaus which is the direct consequence of the transport through the top and bottom Dirac surfaces $\sigma_{xy} = 2(N+1/2)$ and not standard integer quantum Hall effect (IQHE). The higher plateaus are smeared by temperatures, but can be recognized in the derivative of the Hall conductivity in Fig. 3a. The position of the maxima in the derivative are shown in Fig. 3b as a function of the inverse magnetic field, which is extrapolated to $1/2$ indicating a Berry's phase shift of π with respect to ordinary IQHE.

Furthermore, the smallest Faraday rotation in conventional quantum Hall regime is $\theta = 2\alpha$. The factor of 2 comes from the equal contributions of the spin-up and spin-down subsystems which independently exhibit the IQHE. In our case, there is no spin degeneracy because the topological

surface states are spin-momentum locked. Therefore, our result $\theta = \alpha$ is qualitatively different from that expected for a conventional 2D electron gas with a parabolic dispersion (as well as for graphene) in strong magnetic fields. This issue is discussed in detail on the first page of the manuscript.

Comment 3: “Temperatures of these experiments should be noted in all Figures.”

The bath temperature of 1.85 K is indicated in Fig. 1.

Comment 4: “Authors assumed the heating of the electronic system. The power of terahertz beam is several tenth micro W, it seems that the energy flux is low enough to avoid the heating to 25 K. In addition, the transparency in Fig. 1 is high.”

The electron temperature can be higher than the lattice temperature, as discussed in Ref. 35. Generally, we assume that heating of the electron system is only one of the possible explanations. It is explicitly mentioned page 10: “Another explanation is based on spatial fluctuations of the surface carrier densities, which are likely to occur in our samples due to their large lateral sizes compared to the typical Hall bars used in the dc measurements.”

General Response

We appreciate the recognition by the all three reviewers the quality of our work and that reviewers #1 and #2 recommend publication in the present form

Response to Reviewer #3

Comment 1: "The authors have improved their manuscript, but I am not convinced that it is sufficiently innovative to merit publication in Nature communications. At least in the finite magnetic field, the rotation angle is inevitably suffered from the tail of the classical cyclotron resonance, even if their contribution decreases in the high magnetic field. They concluded that the observed rotation angle is "equal to" the fine structure constant, whereas the error bar (~ 4 mrad) is as large as the quantized rotation angle (~ 7.3 mrad) (Fig. 4c). The current results are still far from the metrological definition of the fine structure constant, while I agree that the observed Faraday rotation indicates the TME. Two groups have already published the similar results in Science and Nature communications, which also indicate the TME, so that the justification for the publication will be necessary. I think the merits of this work are high quality HgTe film and the terahertz spectroscopy with use of gate control. But, in my opinion, these merits do not meet the criteria of the nature communications."

We would like to emphasize that our work was initially posted in arXiv independently and nearly simultaneously with the arXiv papers of Okada et al. and Wu et al., i.e., before they were published in Nature Communications and Science. Furthermore, in contrast to these works, we use transparent gate electrodes, which is a major breakthrough in identifying the TME because it allows an electric control of the surface Landau levels, suppressing the nontopological contribution of the classical cyclotron resonance at a fixed magnetic field.